# Breeding of High Cooking and Eating Quality in Rice by Marker-Assisted Backcrossing (MABc) Using KASP Markers

**DOI:** 10.3390/plants10040804

**Published:** 2021-04-19

**Authors:** Me-Sun Kim, Ju-Young Yang, Ju-Kyung Yu, Yi Lee, Yong-Jin Park, Kwon-Kyoo Kang, Yong-Gu Cho

**Affiliations:** 1College of Agriculture and Life & Environment Sciences, Chungbuk National University, Cheongju 28644, Korea; kimms0121@cbnu.ac.kr (M.-S.K.); yangjy@cbnu.ac.kr (J.-Y.Y.); leeyi22@cbnu.ac.kr (Y.L.); 2Syngenta Crop Protection LLC, Seeds Research, 9 Davis Dr. Research Triangle Park, Durham, NC 27709, USA; yjk0830@hotmail.com; 3College of Industrial Science, Kongju National University, Yesan 32439, Korea; yjpark@kongju.ac.kr; 4Division of Horticultural Biotechnology, Hankyong National University, Anseong 17579, Korea

**Keywords:** rice, cooking and eating quality, recurrent parent, genetic background recovery, molecular breeding, MABc, SNP

## Abstract

The primary goals of rice breeding programs are grain quality and yield potential improvement. With the high demand for rice varieties of premium cooking and eating quality, we developed low-amylose content breeding lines crossed with Samgwang and Milkyqueen through the marker-assisted backcross (MABc) breeding program. Trait markers of the SSIIIa gene referring to low-amylose content were identified through an SNP mapping activity, and the markers were applied to select favorable lines for a foreground selection. To rapidly recover the genetic background of Samgwang (recurrent parent genome, RPG), 386 genome-wide markers were used to select BC_1_F_1_ and BC_2_F_1_ individuals. Seven BC_2_F_1_ lines with targeted traits were selected, and the genetic background recovery range varied within 97.4–99.1% of RPG. The amylose content of the selected BC_2_F_2_ grains ranged from 12.4–16.8%. We demonstrated the MABc using a trait and genome-wide markers, allowing us to efficiently select lines of a target trait and reduce the breeding cycle effectively. In addition, the BC_2_F_2_ lines confirmed by molecular markers in this study can be utilized as parental lines for subsequent breeding programs of high-quality rice for cooking and eating.

## 1. Introduction

Rice (*Oryza sativa* L.) is a vital worldwide agricultural product. It is one of the world’s leading staple crops, as more than half of the world’s population relies on the significant source of calories and protein daily [1]. Indeed, the demand for the improved eating quality of rice is continuously increasing around the globe because it is the most important factor in determining the market price [2]. The improvement of grain quality and yield potential is the primary goal of rice breeding programs. The main components of rice grain quality include appearance, eating, cooking, milling, and nutritional qualities. These rice grain values are determined by their physical–chemical properties and other socio-cultural factors [3].

Milled rice is composed of protein (6.6–7.1%), crude fat (0.3–0.5%), crude fiber (0.2–0.5%), starch (80–82%), and moisture content (12–13%), with starch being the major component of Japonica rice. Starch is a polymeric carbohydrate in which glycoside bonds link a large number of glucose units. In particular, starch in rice consists of amylose and amylopectin [4]. The physical and chemical characteristics of starch and the taste of milled rice varieties depend on the amylose and amylopectin ratio [5]. Rice with a higher amylose content has a coarse, dry texture after cooking, whereas rice with a higher amylopectin content is shiny, sticky, and soft [6,7]. Amylose is synthesized by granule-bound starch synthase I (GBSSI) that is responsible for the biosynthesis of extra-long branch chains of amylopectin, synthesized by soluble starch synthases (SSs), starch-branching enzymes (SBEs), and starch-debranching enzymes (DBEs) [4,8,9]. There are 10 SS isoenzymes (SSI, SSIIa, SSIIb, SSIIc, SSIIIa, SSIIIb, SSIVa, SSIVb, GBSSI, GBSSII) in rice [10,11]. According to Fujita [12], SSIIIa-deficient rice mutants showed that B2 to B4 chains with an amylopectin polymerization degree of 30 or higher were reduced by 60% compared to wild-type rice and that amylose content and the amount of extra-long chains with a polymerization degree of 500 or more were increased by 1.3- and 12-fold, respectively. In addition, it was reported that the structure of starch granules was small and round-shaped, with less crystallization.

Recent research on rice cooking and eating quality has focused on understanding the genetics and genomics of starch biosynthesis and variations such as a starch branching enzyme, wx gene mutation, and soluble starch synthase III genes. Cooking and eating quality is primarily determined by four main physicochemical properties: amylose content (AC), gel consistency (GC), gelatinization temperature (GT), and protein content (PC) [13]. AC is widely demonstrated to be the most crucial factor affecting cooking and eating characteristics and is negatively related to taste palatability, including transparency, viscosity, and rice milling quality [14]. GT is positively correlated with physical characteristics responsible for cooking time and the ability to absorb water during cooking. Rice grains with high GT require more water and cooking time than rice with low or medium GT [7]. GC is an index used to distinguish the cooked texture of high-amylose rice varieties and is analyzed by measuring the cold paste viscosity of cooked milled rice flour. Rice with a soft gel consistency is highly preferred among consumers [13]. Protein content (PC) is negatively correlated to peak viscosity (PV) and hot paste viscosity. Reducing protein content (PC) in rice may increase its peak viscosity [15].

Molecular breeding, such as marker-assisted selection (MAS), could overcome conventional breeding limitations and allows the pyramiding of multiple valuable genes into a single cultivar rapidly and efficiently [16,17]. A marker-assisted backcross selection can recover up to 99% of the recurrent-parent genome in just three backcross cycles, whereas conventional breeding may take up to six backcrossings [18]. The first step of MABc (marker-assisted backcrossing) is selecting individual plants with alleles of the target gene from a donor parent [19,20,21]. In this step, applying trait markers that are tightly linked to the target gene is essential. Otherwise, recombination occurs between markers and target genes, which happens when the markers are no longer associated with the target gene, resulting in selecting untargeted genes [20,22]. The second step is recovering the genetic background of a recurrent parent using genome-wide markers. Two to three markers per 100 cM in an early generation are recommended to conduct rapid genetic background recovery. Applying markers that are well distributed over the whole genome is another key for the success of genetic background recovery [21]. In this study, a Kompetitive allele-specific PCR (KASP) marker was used, which is a polymerase chain reaction-based (PCR) technology using fluorescence for single nucleotide polymorphism (SNP) and small insertion and deletion (InDel). KASP markers have the advantage of a low error rate and a relatively low cost compared to other SNP genotyping platforms such as TaqMan systems [23]. KASP markers are used for various crop breeding, such as pigeon bean [24], chickpea [25], and Indica rice [26]. In Korea, KASP markers were used in the genetic relationship analysis of Korean rice varieties [27] and various quantitative trait loss (QTL) mapping studies such as pre-harvest sprouting resistance and selecting resistant varieties [28].

The objectives of this study were (1) to detect an SNP for a foreground selection for the backcross population by comparing the genome sequences between Samgwang (18.1% AC) and Milkyqueen (12.2% AC), and (2) to develop breeding lines with low amylose content by applying MABc breeding programs to backcross populations between Samgwang and Milkyqueen using KASP markers.

## 2. Results

### 2.1. Evaluation of Characteristics Related to Cooking and Eating Quality in Parents

To evaluate the cooking and eating quality of the Samgwang and Milkyqueen parent plants, we investigated amylose content, protein content, moisture content, whiteness, cooking and texture characteristics, and viscosity characteristics. Koshihikari and Baegjinju, were used as check varieties, because Koshihikari, as a non-glutinous check variety that is known as one of the good cooking and eating varieties [29], was used to compare with Samgwang, and Baegjinju as a semi-glutinous check variety that was one of a similar variety with a semi-glutinous trait of Milkyqueen [30] was used to compare with Milkyqueen. The results are described as follows.

#### 2.1.1. Characteristics Related to Cooking and Texture

The cooking and texture characteristics of Samgwang, Koshihikari, Milkyqueen, and Baegjinju are shown in Table 1. Hardness, a desirable attribute to compress rice when chewing, was evaluated. Samgwang (36.03) and Koshihikari (31.75) were slightly harder than Baegjinju (24.07) and Milkyqueen (21.17). This result demonstrates that Milkyqueen has a lower hardness than Baegjinju and will be softer when chewing. Adhesiveness, which refers to the tenacity of rice, was evaluated. The adhesiveness of Koshihikari (63.44) was higher than that of Samgwang (55.74), and the adhesiveness of Baegjinju (60.94) was higher than that of Milkyqueen (57.90). Springiness is how well a rice grain will physically spring back to its original shape after being deformed by partial compression [31]. The springiness of Koshihikari (34.56) was slightly higher than that of Samgwang (30.92), and the springiness of Baegjinju (26.35) and Milkyqueen (26.21) were similar. Regarding stickiness, Koshihikari (69.56) was higher than Samgwang (48.01), and Baegjinju (76.10) was higher than Milkyqueen (60.03).

#### 2.1.2. Characteristics Related to Eating Quality

The eating quality-related characteristics of Koshihikari, Samgwang, Baegjinju, and Milkyqueen, such as amylose content, protein content, moisture content, and whiteness, are shown in Figure 1. Samgwang (18.1%) and Koshihikari (18.0%) had a similar amylose contents, which was within the amylose content range for good cooking (18–20%) of non-glutinous rice. Meanwhile, Milkyqueen was much lower, at 12.2%, and Baegjinju (9.1%) was even lower than that of Milkyqueen, showing that the amylose content range is around 9–10% for good cooking semi-glutinous rice [32]. The protein content of rice is known to be inversely proportional to the taste of rice and proportional to the viscosity when cooking. In addition, the higher the protein content, the more transparent and harder rice grains are, so cooking needs more water and time [33]. The protein content of the four varieties was similar to the range of Japonica rice (5.91–7.89%), but the protein contents of Milkyqueen (6.3%) and Baegjinju (6.7%) were higher than those of Koshihikari (5.2%) and Samgwang (5.6%) [34]. The moisture content ranged from 10.5 to 14.5%, with an average of 12.2%. Whiteness is an index indicating the whiteness of rice grains, and Koshihikari and Milkyqueen showed similarly with higher values of 44.6% and 43.9%, respectively, than Samgwang (31.5%) and Baegjinju (22.7%).

#### 2.1.3. Viscosity Characteristics by RVA

When starch is heated above gelatinization temperature with sufficient water, the starch particles absorb water and swell several tens of times, increasing in volume and viscosity [35]. Since the gelatinization temperature has a positive correlation with the cooking time, rice varieties with a high gelatinization temperature take more water and a longer time to cook. A low or medium gelatinization temperature is required for high-quality rice varieties, and Japonica rice consumers prefer varieties with a low gelatinization temperature [36]. Koshihikari (72.78) and Samgwang (72.24) have similar amylose contents and similar gelatinization temperatures (Table 2). Baegjinju (69.43) and Milkyqueen (71.43) have lower gelatinization temperatures compared to Koshihikari (72.78) and Samgwang (Table 2). This result is aligned with the previous studies [37,38] that non-glutinous rice has a higher content of amylose, a denser starch structure, and a higher gelatinization initiation temperature than glutinous rice.

Peak viscosity between heating and holding cycles refers to the moisture capacity of starch, and the higher the peak viscosity, the weaker the gelatin that is formed [39]. Koshihikari (392.42) showed a higher peak viscosity than Samgwang (269.08), which means that Koshihikari is softer when it is gelatinized (Table 2), and it is different in hardness in cooking and texture characteristics. Baegjinju (172.78) showed a lower peak viscosity than Milkyqueen (299.15), showing the difference in hardness in cooking and texture characteristics. If heating is continued after reaching peak viscosity, amylose in the starch particles is eluted and broken, resulting in a decrease in viscosity, and hot paste viscosity (minimum viscosity after peak) is reached. Breakdown viscosity (peak viscosity minus hot paste viscosity) is associated with the stability of processing. The lower the breakdown viscosity value, the stronger the resistance to heat and gel breaking, and it is better for processing. The different varieties were ranked by breakdown viscosity: Koshihikari (227.04), Baegjinju (127.25), Samgwang (112.59), and Milkyqueen (90.32), indicating that Milkyqueen was the most favorable for processing.

Cooling gelatinized starch by heating causes amylose molecules to become eluted and entangled and then form a delusional structure. The space is filled with broken starch particles, strengthening the structure and increasing the viscosity [40]. The maximum viscosity is called a cool-paste viscosity. Therefore, the higher the amylose content, the faster the structure is formed, and the cool paste viscosity increases [41]. Consistent with the difference in amylose content, Koshihikari (265.92) and Samgwang (260.97) showed similar cool-paste viscosity, and Baegjinju (77.64) showed a low cool-paste viscosity (Table 2). However, Milkyqueen (294.33) offers the highest cool-paste viscosity despite being a middle glutinous rice, suggesting that aging is important after cooking. Setback viscosity is a value obtained by subtracting the peak viscosity from the cool-paste viscosity. It means that the higher the setback viscosity value, the faster the starch’s aging occurs, which is a positive relationship with amylose content like the cool-paste viscosity. The order of lower setback viscosity is Koshihikari (136.92), Baegjinju (−95.14), Samgwang (19.75), and Milkyqueen (14.42), indicating that Koshihikari is the slowest and Milkyqueen is the fastest to age after cooking (Table 2).

### 2.2. Whole-Genome Re-sequencing Analysis and Variant Discovery for Foreground Selection

We mapped re-sequencing data to SNP to discover genes of interest associated with cooking and eating quality through the genome comparison of Samgwang and Milkyqueen (Appendix A). Milkyqueen is a variety grown by mutagenesis through the treatment of N-methyl-N-nitrosourea (MNU) in Koshihikari. Sequencing reads of selection and filtering were performed on paired-end sequencing reads of Samgwang, Milkyqueen, and Koshihikari according to variants calling pipeline for genome reassembly. As a result, 125,623, 108,446, and 140,887 homozygosity SNPs were identified from the genomes of Samgwang, Milkyqueen, and Koshihikari, respectively. As a result of SNP mapping for each chromosome, the mutation site was identified in the lower part of chromosome 8, consisting of 747 genes. Among them, 706 genes were identified as functional genes by an ORF Finder and assessing amino acid similarity, and SNPs were identified in the CDS region of 268 genes. SNPs responsible for amino acid mutations and trait variations were identified in 135 genes (Appendix A). The SSIIIa gene responsible for the starch synthesis metabolism was discovered in this SNP mapping, and this gene was used for the foreground selection of the backcross breeding process.

#### 2.2.1. Foreground Selection in BC_1_F_1_ and BC_2_F_1_

Next-generation sequencing (NGS) was performed to select functional SNPs in the foreground selection in BC_1_F_1_ and BC_2_F_1_ populations. The SSIIIa gene structure, the location of identified SNPs, and the amino acid variation are shown in Figure 2. To verify the SNP, PCR was performed twice prior to NGS analysis. After performing PCR by synthesizing the first primer with a size of 661 bp based on the location of the target SNP in the SSIIIa gene, a second primer with a size of 202 bp at the target SNP location was prepared using the first PCR product, and PCR was carried out. Table 1 shows the designed first and second primers’ information. As a result of performing NGS analysis of BC1F1 (210) and BC2F1 (196) individuals, it confirmed that the separation ratio of homozygote (G/G): heterozygote (G/A) was 1:1 in BC_1_F_1_ and BC_2_F_1_ generations as expected (Table 3). The SNP genotype in the starch synthase IIIa gene was identified through SNP mapping between high amylose content varieties (G/G genotype): Samgwang (18.2%), Koshhikari (18.1%), Gopum (19.6%), Dongjin (20.3%), Ilpum (18.9%); and low amylose content varieties (A/A genotype): Milkyqueen (12.9%), Baegjinju (9.1%), Seolbaek (10.0%), Josaengheugchal (5.4%), Hangangchal1 (5.0%). The varieties with an A/A genotype revealed low amylose contents (5.0~12.9%) compared with those with high amylose content (18.1~20.3%) varieties. With foreground selection using the SNP of the SSIIIa gene, 93 individuals were selected, followed by recurrent parent genetic background selection.

#### 2.2.2. Background Selection in BC_1_F_1_ and BC_2_F_1_ Generation

A total of 773 KASP markers were developed to discern Japonica rice varieties (Appendix A) and were evaluated to identify the polymorphic markers of Samgwang and Milkyqueen for further background selection. A total of 368 KASP markers were selected, excluding markers that were not amplified or showed hetero genotypes. We applied the second round of marker selection to identify SNPs that were evenly distributed across 12 chromosomes at about 5 Mb intervals within a chromosome resulting in 96 KASP markers (Figure 3). The comprehensive information of 96 KASP markers is provided in Appendix A. Among the lines with recurrent parent genome recovery ratios ranging from 84 to 89% (Figure 4A, Appendix A), seven lines with favorable phenotypes were selected in BC_1_F_1_ (Figure 4B). Six lines (beside SM-46) were similar to the agronomic traits of Samgwang, a recurrent parent (Table 4). The selected BC_1_F_1_ lines were backcrossed with Samgwang to generate the BC_2_F_1_ population. Using the same 96 KASP markers, genotyping of the BC_2_F_1_ population was conducted. The result of the genome recovery ratio of the recurrent parent is shown in Figure 5A. We identified 17 lines with a 90–93% recovery ratio, 19 for 93–95%, 25 for 95–97%, and 25 for 97–99% in BC_2_F_1_ (Appendix A). Seven lines with favorable phenotypes and recurrent parent genome recovery in the range of 97.4–98.9% were selected as candidate lines for high cooking and eating quality (Figure 5B).

### 2.3. Agronomic Traits of the Selected Lines in the BC_2_F_2_ Generation

Seventy seeds of the selected lines (G/A) in BC_2_F_1_ were analyzed and the segregation of P_1_ (G/G): heterozygote (G/A): P_2_ (A/A) was detected as the segregation ratio of 1 (15): 2 (39): 1 (16) by a χ2-test (*p* = 0.05) as expected in BC_2_F_2_ (Appendix A).

The agronomic traits such as plant height, culm length, panicle length, and the number of tillers of the selected individuals in BC_2_F_2_ generation were investigated, as shown in Table 5. Plant height ranged from 100.2 to 115.2 cm, similar to the recurrent parent (112.3 cm). Culm length was 84.2 to 88.8 cm, similar to the recurrent parent (85.6 cm). Therefore, it confirmed that the individuals with a high genome recovery rate selected by KASP markers correlated well with phenotypes of the recurrent parent, Samgwang. The amylose content of Samgwang was 18.5%, and Milkyqueen was 12.6%. The seven selected BC_2_F_2_ seeds showed a distribution of amylose content range of 12.4–16.8% (Table 5). As a result, it confirmed that a breeding material with the favorable trait (low-amylose content) having a high genome recovery ratio of the recurrent parent could be developed by a marker-assisted backcross breeding process efficiently (MABc).

### 2.4. Viscosity Properties of the Selected Lines in BC_2_F_2_ Generation

Viscosity characteristics were investigated for 15 lines in which Samgwang’s genome recovery rate was 97% or higher with the target SNP of the SSIIIa gene derived from Milkyqueen (Table 6). The gelatinization initiation temperature of 15 lines was similar to each other, with the highest viscosity between Samgwang and Milkyqueen. The breakdown viscosity of all lines was higher than Milkyqueen. The final viscosity was also lower than Milkyqueen. The setback viscosity was lower than Milkyqueen but showed a tendency to be higher than Samgwang. Interestingly, the SM35-3 line showed a medium amylose content with low gelatinization initiation temperature but a low peak and breakdown viscosity. Moreover, since the final viscosity is low, the aging of starch is unlikely to happen. The SM35-8 line has the highest breakdown viscosity and the lowest setback viscosity, explaining that the processing stability is high and the aging of starch is slow. This line will be useful for developing high cooking and eating quality rice varieties. According to the analysis of the viscosity characteristics of the selected systems, it demonstrates that the highest viscosity is higher than or similar to that of Samgwang, and the final and setback viscosities are similar to or lower than that of Samgwang so that the texture is smooth, aging does not appear well, and progress is slow. This result reconfirmed that MABc is the recommended breeding technology to develop breeding lines efficiently and effectively.

## 3. Discussion

Rice is the world’s most essential food resource and the smallest genome size among crop species such as corn, soybean, and wheat. It has far superior genetic and breeding research achievements, making it a suitable crop to conduct genetic research. Research on the quality improvement of rice is mainly focused on improving the composition of amylose and amylopectin [42], amylose content [43], gel consistency [44], and gelatinization temperature [45] in Japonica rice. However, studies on the genes responsible for quality and taste in the genomic analysis are rarely conducted. Conventional breeding technology is not suitable for quality improvement in rice; thus, new molecular breeding technology is essential to develop high-quality rice varieties. In this study, we demonstrated the process of MABc using KASP markers by selecting low amylose content lines to recover the genetic background of the recurrent parent as quickly and efficiently as possible.

The starch properties rely on the physicochemical composition of amylose and amylopectin, which play an important role in determining the cooking and eating quality. SSIII is a soluble starch synthesis enzyme, and in SSIIIa, it is mainly expressed in endosperm and is responsible for the synthesis of long-chain amylopectin [46]. Previous studies have found that the content of long-chain amylopectin with polymerization of 30 or more decreased by about 60% after knock-out SSIIIa and has a slight effect on GT [12]. SSIIIa has been reported to affect AC in different populations [47] and contribute to the synthesis of short-chain, the elongation of A and B1 chains, and the formation of B1 and B2 long chains of amylopectin [48]. Studies using mutant or anti-sense inhibition showed that SSIIIa was involved in determining the rice quality and endosperm amylopectin structure, respectively [49]. Therefore, we used the SSIIIa as our target gene to identify the SNP marker for foreground selection for cooking and eating quality improvement of middle-glutinous rice through the SNP mapping of Samgwang and Milkyqueen. In this study, we have considered that the parents and the check varieties showed variations in amylose content by high AC (GG) and low AC (AA) types, this SNP marker can be used to select the amylose content with wx for the foreground selection. The line selection based on the genotype of KASP markers were successful in the BC_1_F_1_ and BC_2_F_1_ generations with the recovery ratios of recurrent parent genome (RPG) in the range of 97~99.1%. We adapted SNP markers to shorten the breeding cycle for background selection, and KASP markers were our technology choice. SNPs for KASP markers associated with cooking and eating quality were discovered through whole-genome re-sequencing analysis followed by genetic mapping. Once we developed SNP markers, the applications of these markers can be extended to other purposes such as distinguishing genetically close lines, mapping agronomically favorable genes, and analyzing germplasm diversity, etc.

Marker-assisted backcross (MABc) is a breeding technique that shortens the breeding generations it takes for offspring obtained using backcross to recover to recurrent parents through selection using molecular markers [50]. The conventional backcross process takes more than six or seven generations, but MABc can reduce the breeding cycle by shortening it to at least three or four generations [20]. MABc is selected using molecular markers (foreground selection), not only the progeny with the donor parent’s heterozygous properties but also the backcross generations with a high genome recovery ratio of the recurrent parent, using several chromosomal-specific molecular markers (background selection). It has been successfully applied to various crops, including rice, soybeans, rye, and others, and contributes to developing new varieties successfully [51]. Recent advances in single-plex single nucleotide polymorphism (SNP) genotyping techniques such as TaqMan, SimpleProbe, and Kompetitive Allele-Specific PCR have enabled MABc to use functional SNPs. Currently, a KASP (Kompetitive Allele-Specific PCR) assay is being used in various crops due to its low cost and locus specificity and efficiency [52]. The KASP assay is based on fluorescence resonance energy transfer (FRET) in the SNP genotype of allele-specific oligo elongation [53].

In this study, out of 773 KASP markers tested in two parents, 368 polymorphic markers were identified. For background selection analysis, out of 368 KASP markers, 96 KASP markers were selected at about 5 Mb intervals per chromosome, which were evenly distributed on 12 rice chromosomes. The recovery ratio of the recurrent parent genome (RPG) was analyzed using these 96 markers. Twenty-five lines were selected, which were in the range of 97–99.1% of the RPG recovery ratio, and eventually, seven lines with favorable phenotypes were selected in BC_2_F_1_. In the investigation of agricultural traits using these markers, it was judged that most traits recovered the Samgwang traits well due to the high genome recovery ratio through backcrossing between Samgwang and Milkyqueen, resulting in a degree similar to those of the parent variety of Samgwang. A distribution of amylose content was revealed in the BC_2_F_2_ seeds from the selected lines, with a range of 12.4–16.8%, implying that the selection by MABc in this study can be used as additional breeding materials for developing high eating quality rice varieties. Since this study performed MABc on BC_1_F_1_ and BC_2_F_1_ plants in the early generations of backcrossing, the target individuals were selected in a relatively short breeding cycle. We believe the lines of starch mutants for high cooking and eating quality can be used as new breeding materials to develop high-quality rice varieties for eating in any rice breeding program.

## 4. Materials and Methods

### Plant Materials

Two Korean rice varieties were used, Samgwang [54] and Milkyqueen [55], for developing a backcross population, and two varieties, Koshihikari and Baegjinju, were used as check varieties. They were cultivated and harvested according to the standard cultivation method of the Rural Development Administration (RDA) in Korea [56] in an experimental paddy field at Chungbuk National University in 2019–2020. Seeds were dried until a moisture content of 13% was achieved, and the hulls were removed using a roller husking machine. The brown rice was polished using a polishing machine (MC-90A, Toyoseiki) to a grinding rate degree of 90%. For viscosity analysis, a 100-mesh screen (Cyclotec Sample Mill, Tecator Co., Sweden) was used to make rice flour out of 15 g of milled rice. The analyses of cooking and eating traits were applied with three biological and technical replicates.

## 5. Cooking and Eating Quality Characteristics of Milled Rice

### 5.1. Analysis of Amylose and Protein Contents

Amylose and protein contents were measured in a non-destructive method using the Infratec 1241 Grain Analyzer (FOSS, Hilleröd, Sweden). They were investigated in three replicates for each treatment [57]. Near-infrared spectroscopy is a device that measures the concentration of components such as moisture, protein, fat, and carbohydrates by shooting electromagnetic waves of 780–2500 nm wavelengths into a sample and measuring their absorbance. This instrument has the advantage of being fast and non-destructive and can measure several components simultaneously [58].

### 5.2. Characterizations of Cooking and Eating Texture

Milled rice samples (30 g) were cooked for 30 min and cooled for 25 min using a cooler. Cooked milled rice (10 g) was placed on an experimental plate and used to investigate cooking taste characteristics by a full cup method [59]. A TensiPresser Analyzer (My Boy, TAKETOMO Electric Incorporated, Tokyo, Japan) was used to investigate cooking and eating quality and measure hardness, adhesiveness, springiness, stickiness, and thickness.

### 5.3. Rapid Visco Analysis (RVA)

Rice flour (3 g) was blended with 25 mL of distilled water in an aluminum canister and transferred to a Rapid Viscosity Analyzer (Model RVA-4, Newport Scientific Ltd., Warriewood, Australia). An RVA analysis condition was started at 50 °C for one minute and heated to 95 °C at the rate of 12 °C per minute. The temperature was set to remain at 95 °C for two minutes and cool to 50 °C for seven minutes. The paste viscosity properties were gelatinization temperature (°C), peak viscosity (RVU), hot paste viscosity (RVU), cool paste viscosity (RVU), and breakdown (highest viscosity–lowest viscosity) and setback (cool paste viscosity–highest viscosity) were derived.

## 6. MABc Breeding Process

Samgwang, as a female parent, was crossed with Milkyqueen as a male parent to develop the F1 generation. F1 plants were backcrossed to Samgwang as a recurrent parent to generate BC_1_F_1_ and BC_2_F_1_ generations. KASP markers were applied to each generation for the foreground and background selection. Of the 210 total individuals at BC_1_F_1_, 93 plants were selected through the foreground selection. The genetic background of those 93 individuals was screened, and seven selected individuals were advanced to the BC_2_F_1_ population. At BC_2_F_2_, 15 individuals were selected with the same foreground and background selection performed at the BC_2_F_1_. Eating quality-related traits were measured for these 15 BC_2_F_2_ seeds.

## 7. Molecular Marker Analysis

### 7.1. Semi-Nested PCR Analysis for Foreground Selection

Whole-genome re-sequencing of Samgwang and Milkyqueen was carried out using HiSeq 2500 Sequencing System (Illumina, San Diego, CA, USA), and short-read sequences were aligned using the Bowtie program. Assembly and mapping were performed using CLC Main Workbench Software (QIAGEN, Hilden, Germany). Mutation analysis was performed by comparison with the reference rice genome sequence of the IRSGP1.0 (International Rice Genome Sequencing Project). The foreground selection marker was designed in the range of up to 1000 bp, including the mutation region of the OsSSIII gene (Appendix A). DNA was extracted from leaves of the BC_1_F_1_ and BC_2_F_1_ populations to select individuals with target genotypes, and deep-sequencing was performed based on semi-nested PCR using the method in Chi [60].

### 7.2. KASP Marker Analysis for Background Selection and Investigation of Agronomic Traits

The 773 KASP markers developed for Korean Japonica rice cultivars [27] were used to select lines with a high genome recovery ratio of the recurrent parent for the BC_1_F_1_ and BC_2_F_1_ populations between Samgwang and Milkyqueen. Out of 773 KASP markers, 386 polymorphic KASP markers were identified between two parents using the KASP marker analysis system at the Seed Industry Promotion Center of the Foundation of Agri. Tech. Commercialization and Transfer (FACT) (Gimje, Korea). The markers showing heterozygote genotypes were excluded for further analysis. Ninety-six markers located at a 5 Mb intervals on 12 rice chromosomes were selected to analyze the recovery ratio of the recurrent parent genome in BC_1_F_1_ and BC_2_F_1_. Genetic graphical mapping was performed using a MapChart program (version 2.32) based on the physical location of the KASP markers on chromosomes [61]. The agronomic traits such as plant height, culm length, panicle length, and the number of tillers were phenotyped based on a standard method of RDA, Korea [56] in the lines selected in the BC_1_F_1_ and BC_2_F_1_ generations. In addition, amylose content and viscosity characteristics were investigated as eating quality-related traits using 15 BC_2_F_2_ lines with a recovery rate of 97–99.1% of the recurrent parent genome.

## 8. Statistical Analysis

Statistical analysis of the investigated cooking and eating quality characteristics was performed using the SAS program (SAS Institute Inc., Cary, NC, USA). Basic statistics such as mean and deviation of each trait data were analyzed, and the distribution of variation by trait was investigated. Significance (*p* < 0.05) was tested by analyzing variance, and significant differences were compared and analyzed by performing Duncan’s Multiple Range Test.

## 9. Conclusions

The goals of rice breeding programs are the improvement of grain quality and yield potential. With the high demand for rice varieties of premium cooking and eating quality, we developed low-amylose content breeding lines through the marker-assisted backcross (MABc) breeding program from the cross between Samgwang (18.1% AC) and Milkyqueen (12.2% AC). Seven BC_2_F_1_ lines with targeted traits were selected, and the genetic background recovery range varied within 97.4–99.1% of RPG. The amylose content of the selected BC_2_F_2_ grains ranged from 12.4–16.8%. We demonstrated the MABc using trait and genome-wide markers, allowing us to efficiently select lines of a target trait and reduce the breeding cycle effectively. In addition, the BC_2_F_2_ lines confirmed by molecular markers in this study can be utilized as parental lines for subsequent breeding programs of high-quality rice for cooking and eating.

## Figures and Tables

**Figure 1 plants-10-00804-f001:**
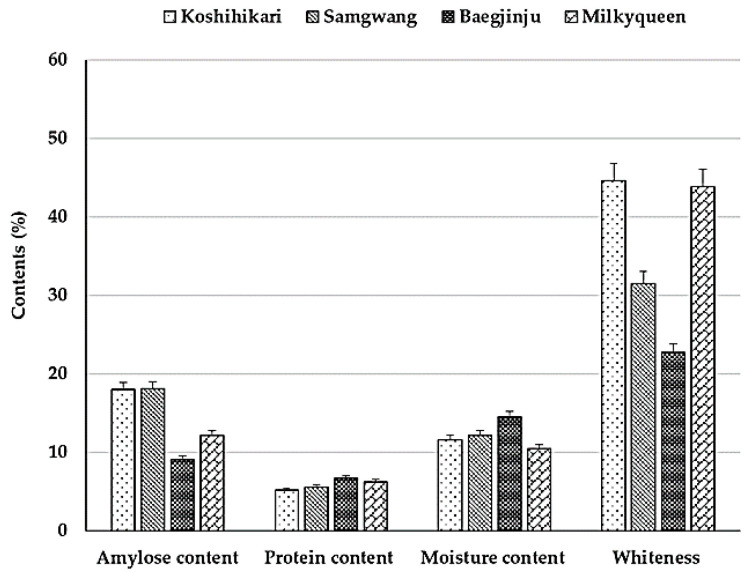
Characteristics related to eating qualities, amylose content (AC), protein content (PC), moisture content and whiteness, were measured in Koshihikari, Samgwang, Baegjinju and Milkyqueen varieties.

**Figure 2 plants-10-00804-f002:**
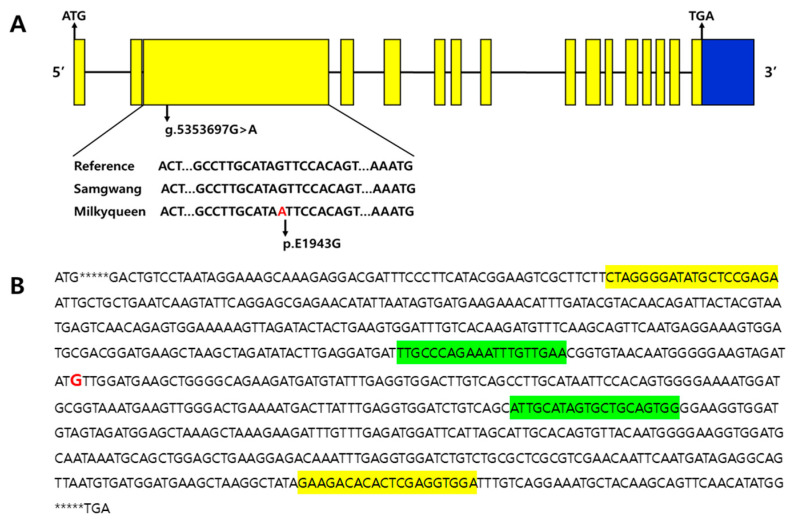
Overview of the SSIIIa gene structure, SNP location and amino acid substitution. (**A**) SNP loci(G/A) revealing missense non-synonymous amino acid substitution (Glutamic acid(E) to Glycine(G): *p*. E1943G) in 3rd exon of the SSIIIa gene. Yellow box: CDS; blue box: downstream. (**B**) Positions of 1st and 2nd PCR primer sets and SNP (red color) used for foreground selection in CDS region of SSIIIa gene. Yellow highlights: 1st PCR primer set; Green highlights: 2nd PCR primer set.

**Figure 3 plants-10-00804-f003:**
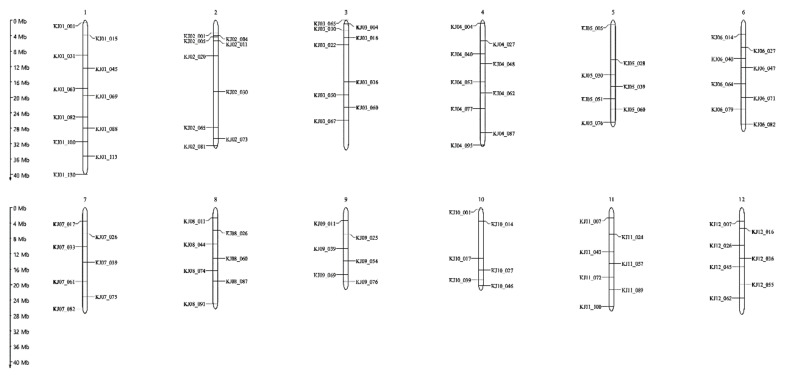
Chromosomal position of 96 KASP markers for background selection that showed SNP polymorphism between Samgwang and Milkyqueen as selected at an interval of about 5 Mb in each chromosome.

**Figure 4 plants-10-00804-f004:**
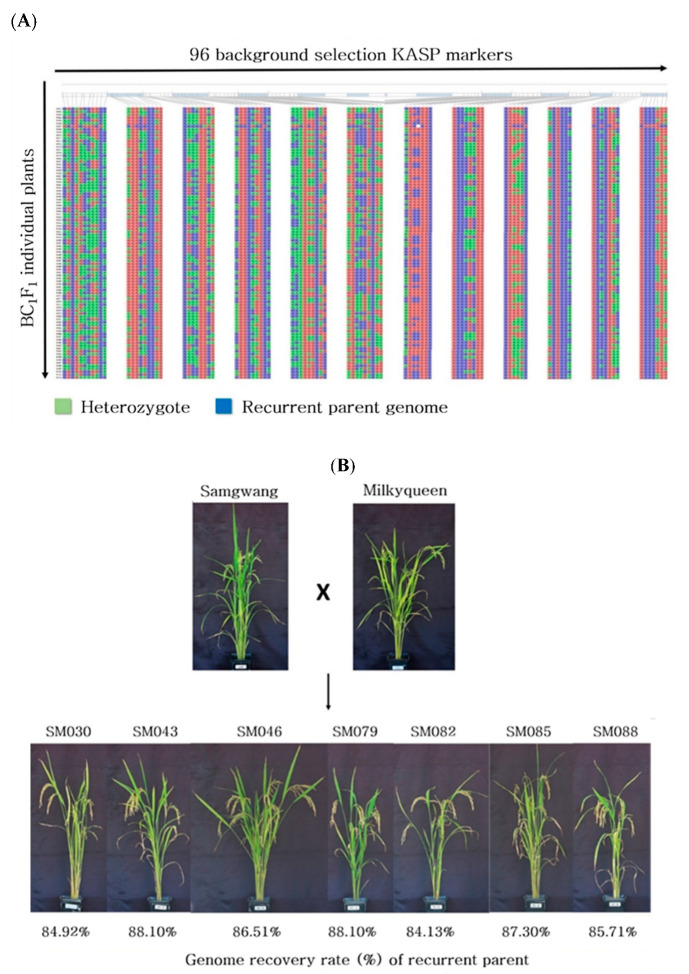
(**A**) Genotyping results with 96 KASP markers for the BC_1_F_1_ population of a total of 93 individuals. (**B**) Phenotype of seven BC_1_F_1_ lines with highest genome recovery rate of the recurrent parent, Samgwang.

**Figure 5 plants-10-00804-f005:**
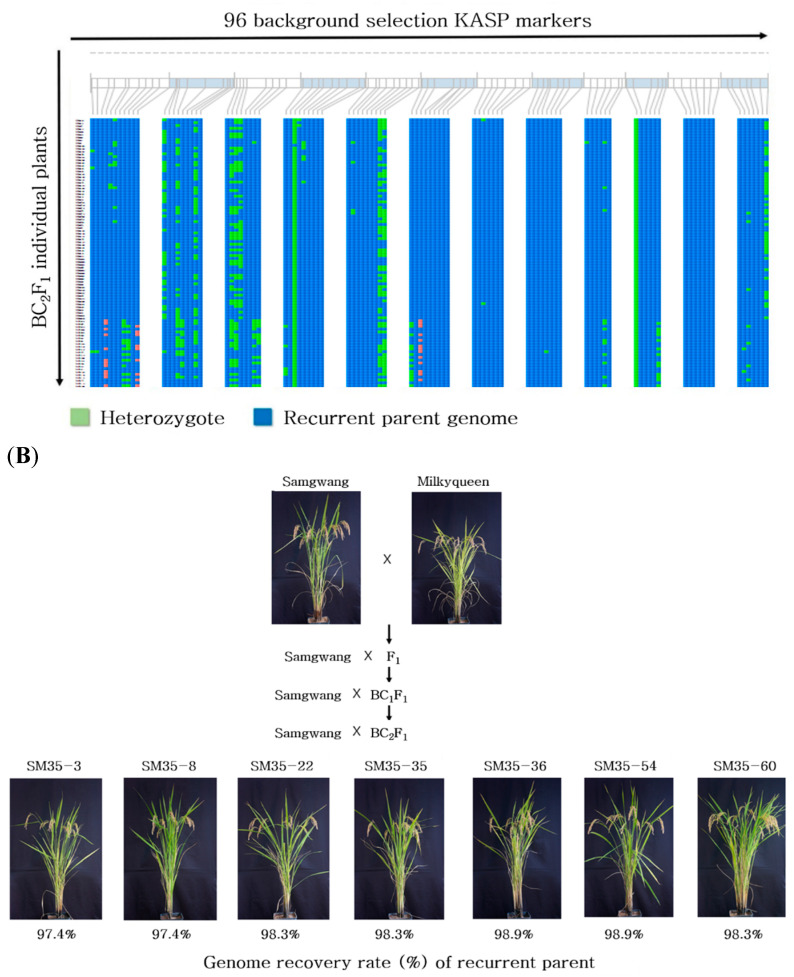
(**A**) Genotyping results of 93 BC_2_F_1_ plants with 96 KASP markers for the background selection in the backcross population between Samgwang and Milkyqueen. (**B**) Phenotype of seven BC_2_F_1_ lines with highest genome recovery rate of the recurrent parent, Samgwang.

**Table 1 plants-10-00804-t001:** Characteristics of texture and eating quality in Koshihikari, Samgwang, Baegjinju and Milkyqueen.

Variety	Hardness	Adhesiveness	Springiness	Stickiness
Koshihikari	31.75 ± 5.73	63.44 ± 2.37	34.56 ± 4.26	69.56 ± 3.49
Samgwang	36.03 ± 0.26	55.74 ± 2.47	30.92 ± 0.94	48.01 ± 3.42
Baegjinju	24.07 ± 3.68	60.94 ± 5.73	26.35 ± 2.95	76.10 ± 3.25
Milkyqueen	21.17 ± 4.36	57.90 ± 3.36	26.21 ± 1.82	60.03 ± 4.85
F-value	8.572 **	1.094	6.233 *	3.099

* *p* < 0.05; ** *p* < 0.01.

**Table 2 plants-10-00804-t002:** Pasting properties in Koshihikari, Samgwang, Baegjinju, and Milkyqueen.

Variety	GT * (°C)	Viscosity (RVU)				
		PV ^§^	HPV ^¶^	Breakdown	CPV ^†^	Setback
Koshihikari	72.78 ± 0.70	392.42 ± 9.90	175.79 ± 15.18	227.04 ± 11.46	265.92 ± 11.71	−136.92 ± 9.44
Samgwang	72.24 ± 0.34	269.08 ± 4.92	168.14 ± 6.74	112.59 ± 18.86	265.98 ± 9.33	−19.75 ± 13.73
Baegjinju	69.43 ± 0.45	172.78 ± 8.46	127.25 ± 5.78	127.25 ± 5.78	77.64 ± 4.64	−95.14 ± 3.97
Milkyqueen	71.43 ± 0.11	299.15 ± 8.68	208.83 ± 8.68	90.32 ± 19.25	294.33 ± 7.95	−14.42 ± 13.27
F-value	38.831 ***	147–951 **	51.268 ***	59.717 ***	365.712 **	108.545 ***

* GT; Gelatinization temperature, ^§^ PV; Peak viscosity, ^¶^ HPV; Hot paste viscosity, ^†^ CPV; Cool paste viscosity. ** *p* < 0.01; *** *p* < 0.001.

**Table 3 plants-10-00804-t003:** Statistical analysis for genotyping results of foreground selection with backcross populations of BC_1_F_1_ and BC_2_F_1_ from the cross between Samgwang and Milkyqueen.

Population.	Generation	Number of Plants			**χ^2^ Value** **(1:1)**
		Total	Homo (G/G)	Hetero (G/A)	
Samgwang × Milkyqueen	BC_1_F_1_	210	110	100	0.48 ^ns^
	BC_2_F_1_	196	94	102	0.33 ^ns^

d.f = 1: χ^2^ (0.05,1) = 3.84. ns: not significant.

**Table 4 plants-10-00804-t004:** Agronomic traits of the selected BC_1_F_1_ lines as compared with parents.

Population	Line	Plant Height (cm)	Culm Length (cm)	Panicle Length (cm)	No. of Tillers
Parent	Samgwang	95.2 ± 4.6	64.8 ± 6.0	19.7 ± 1.5	9 ± 3
	Milkyqueen	92.6 ± 2.9	69.9 ± 5.4	18.4 ± 0.4	10 ± 2
BC_1_F_1_	SM-30	95.8	71.0	22.2	9
	SM-43	93.8	72.4	18.4	7
	SM-46	101.6	72.2	18.4	16
	SM-79	90.2	64.0	20.4	6
	SM-82	91.8	68.0	20.6	7
	SM-85	91.4	71.6	21.8	9
	SM-88	92.6	64.0	20.8	6

**Table 5 plants-10-00804-t005:** Agronomic traits of the selected BC_2_F_2_ lines in comparison with parents.

Population	Line	Plant Height (cm)	Culm Length (cm)	Panicle Length (cm)	No. of Tillers	Amylose Content(%)
Parent	Samgwang	112.3 ± 4.6	85.6 ± 2.4	21.9 ± 1.6	10.6 ± 1.8	18.5 ± 2.1
	Milkyqueen	102.2 ± 3.0	74.6 ± 3.7	16.4 ± 1.1	8 ± 1.2	12.6 ± 1.3
BC_2_F_2_	SM35-3	100.2	83.6	19	10	12.4 ± 5.2
	SM35-8	114.6	85.6	19.8	10	16.8 ± 5.2
	SM35-22	114.2	86	17.6	11	14.2 ± 2.2
	SM35-35	113.6	84.2	18	13	15.9 ± 7.5
	SM35-36	116	82.9	17.4	13	15.1 ± 6.6
	SM35-54	112.2	88.8	17.6	11	14.1 ± 4.7
	SM35-60	115.2	85.6	18.2	14	14.2 ± 4.2

**Table 6 plants-10-00804-t006:** Viscosity properties of the selected BC_2_F_2_ lines in comparison with parents.

	GT* (°C)		Viscosity (RVU)									
Population			PV ^§^		HPV ^¶^		Breakdown		CPV ^†^		Setback	
	Mean ± SD	*t*	Mean ± SD	*t*	Mean ± SD	*t*	Mean ± SD	*t*	Mean ± SD	*t*	Mean ± SD	*t*
Samgwang	72.5 ± 0.4	−6.22 **	228.4 ± 9.5	3.396 **	139.0 ± 9.7	−1.114 ^ns^	94.2 ± 4.9	10.666 ***	210.9 ± 5.7	−5.399 ***	−17.4 ± 3.7	−21.986 ***
Milkyqueen	76.6 ± 1.04		205.9 ± 6.2		146.5 ± 6.1		59.4 ± 2.7		240.4 ± 7.4		34.4 ± 1.5	
SM35-02	70.80		210.25		124.50		85.75		210.42		0.17	
SM35-03	70.73		206.09		125.96		80.13		204.21		−1.88	
SM35-05	71.25		246.42		135.17		111.25		216.63		−29.79	
SM35-08	73.63		249.92		136.38		113.54		217.29		−32.63	
SM35-20	72.83		245.88		137.17		108.71		222.04		−23.84	
SM35-21	72.43		232.00		139.79		92.21		222.80		−9.21	
SM35-31	72.15		241.38		136.50		104.88		215.75		−25.63	
SM35-35	71.53		244.58		142.96		101.63		224.00		−20.59	
SM35-36	72.20		245.25		144.04		101.21		224.00		−21.25	
SM35-39	72.15		225.92		144.50		81.42		231.92		6.00	
SM35-40	72.10		237.25		155.67		81.58		238.17		0.92	
SM35-54	71.93		239.84		140.88		98.96		224.96		−14.88	
SM35-55	72.08		226.50		126.25		100.25		205.04		−21.46	
SM35-60	72.13		228.88		131.88		97.00		210.17		−18.71	
SM35-62	71.40		232.17		139.50		92.67		221.08		−11.08	

* GT; Gelatinization temperature, ^§^ PV; Peak viscosity, ^¶^ HPV; Hot paste viscosity, ^†^ CPV; Cool paste viscosity. ** *p* < 0.01; *** *p* < 0.001; ns, not significant.

## Data Availability

Not applicable.

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
