# Peer review of "Breeding of High Cooking and Eating Quality in Rice by Marker-Assisted Backcrossing (MABc) Using KASP Markers"

_plants, 2021, doi:10.3390/plants10040804_

Round 1
Reviewer 1 Report
In this manuscript, Kim et al reported an example of using marker-assisted backcross strategy to develop low-amylose content breeding lines.
The manuscript is well-structured with adequate introduction and method description. The results support the authors’ conclusion. One suggestion is to soften the acclaim about the identification of genes controlling cooking and eating quality (page 3, line 98). More assays needed to make that acclaim (RNA-Seq analysis in targeted tissue, et al)
- Table 1, in the footnote “∗∗∗ P<0.001”, there is no *** in the table.
- Fig 1. The figure 1 is incomplete. Seems truncated.
- For the single SNP (G/A). The authors need to discuss whether this genotype has been identified in other rice lines (a simple BLAST)
- Beside SNPs, the authors should try to also call structural variations (insertion/deletions) around the chosen markers.
Author Response
Response to Reviewer’s Comments:
We would like to thank the reviewers on valuable comments and suggestions for the manuscript. We believe that the comments have identified important areas that required improvement. The revisions or answers are as follows:
Reviewer 1
In this manuscript, Kim et al reported an example of using marker-assisted backcross strategy to develop low-amylose content breeding lines. The manuscript is well-structured with adequate introduction and method description. The results support the authors’ conclusion.
Comment 1: One suggestion is to soften the acclaim about the identification of genes controlling cooking and eating quality (page 3, line 98). More assays needed to make that acclaim (RNA-Seq analysis in targeted tissue, et al).
Response 1: We have revised the part (page 3, line 98) according to the comments and suggestions. We found that the sentence was not properly written. So we rewrote it as follows: “1) to detect a SNP for a foreground selection for the backcross population by comparing the genome sequences between Samgwang (18.1% AC) and Milkyqueen (12.2% AC),”
Comment 2: Table 1, in the footnote “∗∗∗ P<0.001”, there is no *** in the table.
Response 2: We have deleted “∗∗∗ P<0.001” in Table 1 according to the comments.
Comment 3: Fig 1. The figure 1 is incomplete. Seems truncated.
Response 3: We have modified Figure 1 and renamed the legend as follows: “Characteristics related to eating qualities, amylose content (AC), protein content (PC), moisture content and whiteness, were measured in Koshihikari, Samgwang, Baekjinju and Milkyqueen varieties.”
Comment 4: For the single SNP (G/A). The authors need to discuss whether this genotype has been identified in other rice lines (a simple BLAST).
Response 4: We have included the SNP genotype information on page 7 as follows: “The SNP genotype in the starch synthase IIIa gene was identified through SNP mapping between Samgwang/Koshhikari/Gopum/Dongjin/Ilpum varieties (G/G genotype) and Milkyqueen/Baegjinju/Seolbaek/Josaengheugchal/Hangangchal1 varieties (A/A genotype). The varieties having A/A genotype revealed low amylose contents (5.0~12.9 %) compared to those with high amylose content (ranged 18.1~20.3 %) varieties.”
Comment 5: Beside SNPs, the authors should try to also call structural variations (insertion/deletions) around the chosen markers.
Response 5: We identified SNP mutations in the starch synthase IIIa gene through SNP mapping between Samgwang and Milkyqueen. Among them, SNP (g.5353697G>A), which induces amino acid mutation, was selected as the foreground selection marker. Any insertion /deletion was not detected around the 5Kb region before and after the target SNP of SSIIIa gene.

Reviewer 2 Report
Dear Authors
The present manuscript is organized and well written, although there are still places to improve it. Please find below my suggestions.
1- Figure 1 may be replaced with better quality image, i could not see the complete image and the descriptions.
2- Discussion may be improved and more references need to be discussed in detail in relation to present work.
3-Short conclusion may be incorporated to convey the key finding.
Thank you
Regards
Author Response
Response to Reviewer’s Comments:
We would like to thank the reviewers on valuable comments and suggestions for the manuscript. We believe that the comments have identified important areas that required improvement. The revisions or answers are as follows:
Reviewer 2
The present manuscript is organized and well written, although there are still places to improve it. Please find below my suggestions.
Comment 1: Figure 1 may be replaced with better quality image, i could not see the complete image and the descriptions.
Response 1: We have modified Figure 1 and renamed the legend as follows: “Characteristics related to eating qualities, amylose content (AC), protein content (PC), moisture content and whiteness, were measured in Koshihikari, Samgwang, Baegjinju and Milkyqueen varieties.”
Comment 2: Discussion may be improved and more references need to be discussed in detail in relation to present work.
Response 2: We have improved ‘Discussion’ on page 14, line 385~433, and the conclusion was newly included on page 14~15.
Comment 3: Short conclusion may be incorporated to convey the key finding.
Response 3: We have included the conclusion as follows: “The goals of rice breeding programs are the improvement of grain quality and yield potential. With the high demand for rice varieties of premium cooking and eating quality, we developed low-amylose content breeding lines through the marker-assisted backcross (MABc) breeding program from the cross between Samgwang (18.1% AC) and Milkyqueen (12.2% AC). Seven BC2F1 lines with targeted traits were selected, and the genetic background recovery range varied with 97.4–99.1% of RPG. The amylose content of selected BC2F2 grains was shown a range of 12.4–16.8%. We demonstrated the MABc using trait and genome-wide markers, allowing us to efficiently select lines of accurate genetic components and reduce the breeding cycle effectively. In addition, the BC2F2 lines confirmed by molecular markers in this study can be utilized as parental lines for subsequent breeding programs of high-quality rice for cooking and eating.”

Reviewer 3 Report
The manuscript entitled “Breeding of High Cooking and Eating Quality in Rice by Marker-Assisted Backcrossing (MABc)” by Kim et al, and authors used KASP marker assisted backcrossing for the development of the rice line with enriched cooking and eating quality. I appreciate the effort of the authors. Authors showed the more than 97% The results and the techniques are not provided a new insight by this manuscript. However, I have some queries as follows.
3 using KASP Markers
Characteristics of texture and eating quality, author(s) checked four varieties namely Samgwang and Milkyqueen, Koshihikari and Baekjinju.
I would like to know the basis for the using Koshihikari and Baekjinju as checks? Does these two already had sufficient information?
Line 434: What is the genetic background of the Samgwang (♀) and Milkyqueen (♂)?
Table 1 and 2, how many biological and technical replicates used?
Figure 1 is awful, it must need to replace.
Why there is not any error bar as well in the figure 1, and how many biological and technical replicates used?
2.1.2. Characteristics Related to Eating Quality, should check again and rewrite each parameter properly
Table 3, χ2 value does not looks correct (I guess), please check my assumption wrong, or right?
Line 246-250, missing information to jump 368 to 96 markers, and marker segregation analysis as well.
What about the heterozygosity state in BC2F2 population?
Author Response
Response to Reviewer’s Comments:
We would like to thank the reviewers on valuable comments and suggestions for the manuscript. We believe that the comments have identified important areas that required improvement. The revisions or answers are as follows:
Reviewer 3
The manuscript entitled “Breeding of High Cooking and Eating Quality in Rice by Marker-Assisted Backcrossing (MABc)” by Kim et al, and authors used KASP marker assisted backcrossing for the development of the rice line with enriched cooking and eating quality. I appreciate the effort of the authors. Authors showed the more than 97%. The results and the techniques are not provided a new insight by this manuscript. However, I have some queries as follows.
3 using KASP Markers
Comment 1: Characteristics of texture and eating quality, author(s) checked four varieties namely Samgwang and Milkyqueen, Koshihikari and Baegjinju.
I would like to know the basis for the using Koshihikari and Baegjinju as checks? Does these two already had sufficient information?
Response 1: We have revised the part (page 3, line107~112) according to the comments and suggestions as follows: Koshihikari and Baegjinju, were used as check varieties, because Koshihikari, as a non-glutinous check variety that is known as one of good cooking and eating varieties (Kobayashi et al. 2018), was used to compare with Samgwang, and Baegjinju as a semi-glutinous check variety that was one of the similar variety with a semi-glutinous trait of Milkyqueen (Lee et al. 2016) was used to compare with Milkyqueen.
Comment 2: Line 434: What is the genetic background of the Samgwang (♀) and Milkyqueen (♂)?
Response 2: We have revised as follows: “Two Korean rice varieties were used, Samgwang [55] and Milkyqueen [56], for developing a backcross population, and two varieties, Koshihikari and Baegjinju, were used as check varieties.”
Comment 3: Table 1 and 2, how many biological and technical replicates used?
Response 3: We have described this as follows: “The analyses of cooking and eating traits were applied with three biological and technical replicates” in page 15, line 437~438.
Comment 4: Figure 1 is awful, it must need to replace.
Why there is not any error bar as well in the figure 1, and how many biological and technical replicates used?
Response 4: We have modified Figure 1 and renamed the legend as follows: “Characteristics related to eating qualities, amylose content (AC), protein content (PC), moisture content and whiteness, were measured in Koshihikari, Samgwang, Baegjinju and Milkyqueen varieties.”
Comment 5: 2.1.2. Characteristics Related to Eating Quality, should check again and rewrite each parameter properly.
Response 5: Thank you for the reviewer’s comments. We have rewritten as follows: “Samgwang (18.1%) and Koshihikari (18.0%) had a similar amylose content, which was within the amylose content range for good cooking (18–20%) of non-glutinous rice. Meanwhile, Milkyqueen was much lower as of 12.2%, and Baegjinju (9.1 %) was even lower than that of Milkyqueen, showing the amylose content range of around 9–10% for good cooking semi-glutinous rice [32]. The protein content of rice is known to be inversely proportional to the taste of rice and proportional to the viscosity when cooking. In addition, the higher the protein content, the more transparent and harder rice grains are, so cooking needs more water and time [33]. The protein content of the four varieties was similar to the range of Japonica rice (5.91–7.89%), but the protein contents of Milkyqueen (6.3%) and Baegjinju (6.7%) were higher than those of Koshihikari (5.2%) and Samgwang (5.6%) varieties [34]. The moisture content ranged from 10.5 to 14.5%, with an average of 12.2%. Whiteness is an index indicating the whiteness of rice grains, and Koshihikari and Milkyqueen showed similarly with higher values of 44.6% and 43.9%, respectively, than Samgwang (31.5%) and Baegjinju (22.7%).”
Comment 6: Table 3, χ2 value does not looks correct (I guess), please check my assumption wrong, or right?
Response 6: We have corrected it in the Table 3.
Comment 7: Line 246-250, missing information to jump 368 to 96 markers, and marker segregation analysis as well.
Response 7: We have described about 96 KASP markers in the page 8, lines 267-271 as follows: “A total of 368 KASP markers were selected, excluding markers that were not amplified or showed hetero genotypes. We applied the second round of marker selection to identify SNPs that are evenly distributed across 12 chromosomes at about 5 Mb intervals within a chromosome resulting in 96 KASP markers (Figure 3).”
Comment 8: What about the heterozygosity state in BC2F2 population?
Response 8: The genotype, A/A, is selected in the BC2F2 generation. In the BC2F2 generation, the SNP introduced from P2 will be homo A/A and used as the selection line for the desired experimental results.

Round 2
Reviewer 2 Report
Dear Authors
I agree with the changes and manuscript has been improved significantly.
I do not have any further query.
Thank you
Author Response
Thank you for your kind comments on the authors responses.
Best regards ...
Reviewer 3 Report
Authors have answered all of my queries. Thanks for clear description.
However, in Comment 8
Comment 8: What about the heterozygosity state in BC2F2 population?
Authors response 8: The genotype, A/A, is selected in the BC2F2 generation. In the BC2F2 generation, the SNP introduced from P2 will be homo A/A and used as the selection line for the desired experimental results.
Re-comment: We can understand this point easily. My point is that, it is presumed while your newly developed population support the theory or not. I suggest you to check the level of heterozygosity in BC2F2 and add the data which will make a proof to the presumption.
Author Response
Re-comment: We can understand this point easily. My point is that, it is presumed while your newly developed population support the theory or not. I suggest you to check the level of heterozygosity in BC2F2 and add the data which will make a proof to the presumption.
Response: Thank you for your valuable comments. We have revised in page 11, line 301~303 as follows: “Seventy seeds of the selected lines (G/A) in BC2F1 were analyzed and the segregation of P1 (G/G): heterozygote (G/A): P2 (A/A) was detected as the segregation ratio of 1 (15): 2 (39): 1 (16) by a χ2-test (P= 0.05) as expected in BC2F2 (Supplementary Table 4).” Supplementary Table 4 was also uploaded.
